# Antidiabetic, Antiglycation, and Antioxidant Activities of Ethanolic Seed Extract of *Passiflora edulis* and Piceatannol In Vitro

**DOI:** 10.3390/molecules27134064

**Published:** 2022-06-24

**Authors:** Flávia A. R. dos Santos, Jadriane A. Xavier, Felipe C. da Silva, J. P. Jose Merlin, Marília O. F. Goulart, H. P. Vasantha Rupasinghe

**Affiliations:** 1Institute of Chemistry and Biotechnology, Federal University of Alagoas, Maceio 57072-970, Brazil; flavia.santos@iqb.ufal.br (F.A.R.d.S.); jadrianexavier@iqb.ufal.br (J.A.X.); felipe.silva@iqb.ufal.br (F.C.d.S.); mofg@qui.ufal.br (M.O.F.G.); 2Department of Plant, Food, and Environmental Sciences, Faculty of Agriculture, Dalhousie University, 50 Pictou Road, Truro, NS B2N 5E3, Canada; josemerlinj@dal.ca

**Keywords:** passion fruit, piceatannol, agro-industrial residues, type 2 diabetes, cytotoxicity, seeds

## Abstract

The objective of this work was to investigate the antidiabetic, antiglycation, and antioxidant potentials of ethanolic extract of seeds of Brazilian *Passiflora edulis* fruits (PESE), a major by-product of the juice industry, and piceatannol (PIC), one of the main phytochemicals of PESE. PESE, PIC, and acarbose (ACB) exhibited IC_50_ for alpha-amylase, 32.1 ± 2.7, 85.4 ± 0.7, and 0.4 ± 0.1 µg/mL, respectively, and IC_50_ for alpha-glucosidase, 76.2 ± 1.9, 20.4 ± 7.6, and 252 ± 4.5 µg/mL, respectively. The IC_50_ of PESE, PIC, and sitagliptin (STG) for dipeptidyl-peptidase-4 (DPP-4) was 71.1 ± 2.6, 1137 ± 120, and 0.005 ± 0.001 µg/mL, respectively. PESE and PIC inhibited the formation of advanced glycation end-products (AGE) with IC_50_ of 366 ± 1.9 and 360 ± 9.1 µg/mL for the initial stage and 51.5 ± 1.4 and 67.4 ± 4.6 µg/mL for the intermediate stage of glycation, respectively. Additionally, PESE and PIC inhibited the formation of β-amyloid fibrils in vitro up to 100%. IC_50_ values for 1,1-diphenyl-2-picrylhydrazyl radical (DPPH^•^) scavenging activity of PESE and PIC were 20.4 ± 2.1, and 6.3 ± 1.3 µg/mL, respectively. IC_50_ values for scavenging hypochlorous acid (HOCl) were similar in PESE, PIC, and quercetin (QCT) with values of 1.7 ± 0.3, 1.2 ± 0.5, and 1.9 ± 0.3 µg/mL, respectively. PESE had no cytotoxicity to the human normal bronchial epithelial (BEAS-2B), and alpha mouse liver (AML-12) cells up to 100 and 50 µg/mL, respectively. However, 10 µg/mL of the extract was cytotoxic to non-malignant breast epithelial cells (MCF-10A). PESE and PIC were found to be capable of protecting cultured human cells from the oxidative stress caused by the carcinogen NNKOAc at 100 µM. The in vitro evidence of the inhibition of alpha-amylase, alpha-glucosidase, and DPP-4 enzymes as well as antioxidant and antiglycation activities, warrants further investigation of the antidiabetic potential of *P. edulis* seeds and PIC.

## 1. Introduction

The *Passiflora* (Passifloraceae family) genus has more than 500 species; however, *Passiflora edulis* Sims *f. flavicarpa* O. Degenerer (yellow or sour form passion fruit) and *P. alata* (sweet form) are the two main economically important species [1]. Brazil is currently the leading producer of passion fruit in the world, with 690,364 tons of production in 2020 [2]. Several biological properties of passion fruits have been demonstrated using in vitro and in vivo studies, such as antioxidant (leaf, peel, and seed), analgesic, antidepressant, sedative and anxiolytic-like (leaf), anti-inflammatory (peel and leaf), antimicrobial (seed), anti-hypertensive (peel), hepatoprotective (peel and seed), and antidiabetic (peel, juice, and seed) activities [3]. In Brazil, *P. edulis*, also known as “maracuja”, has been used widely in folk medicine as an anxiolytic, sedative, and analgesic agent [4].

One of the main bioactive compounds of seeds of *P. edulis* is piceatannol (PIC, 3,3′,4′,5-*trans*-tetrahydroxystilbene), an analog of resveratrol. Although the health-promoting effects of PIC have not been studied extensively as resveratrol [5], it has been shown to possess preventive effects concerning cardiovascular diseases and certain cancers [6], being anti-inflammatory [7], photoprotective [8], and as a skin hydration [9] and an antidiabetic agent [10].

Diabetes is one of the significant health problems worldwide, mainly due to its increased prevalence, irreversible complications, and high economic burden [11]. The International Diabetes Federation stated that 537 million people aged 20–79 years in the world or about 10.5% of all adults in this age group were affected by diabetes in 2021. More than 90% of diabetes cases are type 2 diabetes mellitus (T2DM), a metabolic disorder characterized by chronic hyperglycemia due to insufficient production of insulin by the pancreas or by insulin resistance [12].

The present therapeutic drugs used in the management of T2DM are sulfonylureas, biguanides, SGLT2 inhibitors, alpha-glucosidase inhibitors, dipeptidyl peptidase-4 (DPP-4) inhibitors, GLP-1 receptor agonists, thiazolidinediones, amylin analogs, and insulin [13]. However, continuous use of some FDA-approved oral drugs is accompanied by side effects; for instance, acarbose (ACB) leads to abdominal or stomach pain, flatulence, diarrhea, bloating, and cramping [14]. As such, there is a continuous need for safe, natural health products that can be used to prevent and manage T2DM.

One of the strategies to prevent postprandial hyperglycemia in T2DM is inhibiting the catalytic activity of digestive enzymes, which are responsible for catalyzing the hydrolysis of oligosaccharides into glucose, including alpha-amylase and alpha-glucosidase. These enzymes increase glucose absorption and, as a result, increase the glucose level in the bloodstream [15]. Another potential therapeutic approach that lowers blood glucose concentration in T2DM is blocking the action of DPP-4, an enzyme that degrades the hormone incretin, which is secreted into the blood after a meal to stimulate insulin secretion from pancreatic β-cells and to regulate glucose production by the liver [13].

Diabetes has also been considered a risk factor for neurodegenerative diseases such as Alzheimer’s disease (AD) [16,17], which is characterized by the accumulation of extracellular insoluble senile plaque and intracellular neurofibrillary tangles. Hyperglycemia, as an inevitable consequence of increased insulin resistance, leads to the glycation of proteins and, consequently the formation of advanced glycation end products (AGEs) [18]. AGEs, which accumulate over some time and are not frequently detoxified, have been associated with amyloid-based neurodegenerative diseases [17], stabilizing protein aggregates [19]. Furthermore, the interaction of AGEs with AGE receptors (RAGE) causes inflammatory processes and oxidative stress in cells. As RAGE is not specific, it can also interact with other ligands such as β-amyloid peptide (Aβ), forming agglomerates, which lead to increased inflammation, oxidative stress, neuronal dysfunction, with consequent AD aggravation [18].

(Poly)phenol-rich plant extracts and individual (poly)phenols have been extensively studied as an alternative or complement to the current hypoglycemic medicines. In vitro studies have demonstrated that (poly)phenols interact with enzymes and other biological macromolecules. The interaction depends on the composition, molecular weight, and the position of hydroxyl groups of the bioactive compounds [20].

Daily administration of PIC caused a decrease in glucose levels in animal models. For instance, PIC supplementation reduced fasting blood glucose in both *db*/*db* mice and high-fat diet-fed C57BL/6J mice [21]. PIC promoted glucose uptake in the absence of insulin in cultured myotubes; enhanced AMP-activated protein kinase (AMPK) activation and glucose transporter type 4 (GLUT4) translocation to overcome insulin resistance [21]. PIC lowered the blood glucose and improved glucose tolerance in diabetic mice [22]. Therefore, PIC has therapeutic potential for the prevention and improvement of symptoms of diabetes.

However, to our best knowledge, no study has been reported in relation to the seeds of Brazilian *P. edulis* (yellow passion fruit) as an antidiabetic agent through the association of alpha-glucosidase, alpha-amylase, and DPP-4 inhibition mechanisms, not even as an antiglycant in the initial and intermediate stages of glycation and as an inhibitor of fibrillation of proteins subjected to glycation. As such, the present study investigated the potential use of bioactive chemical constituents of passion fruit seeds as a natural health product for managing chronic disorders, primarily T2DM. Total antioxidant capacity, antidiabetic, and antiglycation activities of extracts prepared from seeds of Brazilian *P. edulis* and with PIC were investigated. In addition, the cytotoxicity of passion fruit seed extracts (PESE) was studied using three cell lines.

## 2. Results and Discussion

### 2.1. Antidiabetic Activity

In vitro antidiabetic potential of PESE and PIC was investigated by determining the ability to inhibit pancreatic alpha-amylase, alpha-glucosidase, and DPP-4. In this study, ACB and STG were used for comparison purposes. All tested samples showed activity against alpha-amylase, alpha-glucosidase, and DPP-4 in a concentration-dependent manner (Table 1). For pancreatic alpha-amylase, PESE, PIC, and ACB presented IC_50_ values of 32.1 ± 2.7, 85.4 ± 0.7, and 0.4 ± 0.1 µg/mL, respectively. For yeast alpha-glucosidase, the values of IC_50_ of PESE, PIC, and ACB were 76.2 ± 1.9, 20.4 ± 7.6, and 252 ± 4.5 µg/mL, respectively. IC_50_ values of PESE and PIC were higher when compared to ACB for alpha-amylase but significantly lower (*p* ≤ 0.05) for alpha-glucosidase. Interestingly, passion fruit seed presented an IC_50_ value 3-times lower than PIC in alpha-amylase but 4-fold higher in the alpha-glucosidase assay. Moreover, PIC showed antidiabetic activity in terms of alpha-glucosidase, which was 12-fold higher than ACB. For DPP-4 human recombinant enzyme, the values of IC_50_ of PESE, PIC, and STG were, respectively, 71.1 ± 2.6, 1137 ± 120, and 0.005 ± 0.001 µg/mL. The results indicate that other bioactive compounds are also involved in the inhibition of alpha-amylase, while PIC may be a key component against alpha-glucosidase.

The extract of passion fruit peel also displayed higher potential towards alpha-glucosidase than alpha-amylase. The ethanolic extract of a hybrid passion fruit peel was shown to have a weak inhibitory effect in alpha-amylase with IC_50_ values of 1.8 ± 0.1 mg/mL (1800 ± 100 µg/mL), but a stronger effect in alpha-glucosidase with IC_50_ values of 0.6 ± 0.1 mg/mL (600 ± 100 µg/mL), which was 4.3-fold higher than standard drug ACB, with IC_50_ values of 0.3 ± 0.1 mg/mL (300 ± 100 µg/mL) [23].

The same pattern was described by Loizzo et al. [24], when investigating the hypoglycemic activity of ethanol extracts of seeds, peel, and pulp of ten Columbia native *Passiflora* species. Among all studied extracts, *P. ligularis* seeds + pulp had the greatest activity with IC_50_ values of 22.6 and 24.8 µg/mL against alpha-amylase and alpha-glucosidase, respectively, followed by *P. pinnatistipula* (IC_50_ values of 46.4 and 37.7 µg/mL against alpha-amylase and alpha-glucosidase, respectively). Positive control, ACB, had IC_50_ values of 50 µg/mL using starch as the substrate for alpha-amylase and 35.5 µg/mL, when using maltose as the substrate for alpha-glucosidase.

Pan et al. [25] obtained a different IC_50_ value for PIC, isolated from ethanol extract of *P. edulis* Sims seeds, which may be due to different conditions performed (substrate, enzyme concentrations, and incubation time). However, PIC derivatives exhibited potent inhibition against in vitro alpha-glucosidase with IC_50_ values ranging from 1.7 to 35.1 μM. Among them, IC_50_ of PIC was 4.3 ± 0.07 μM, while for ACB, it was 218 ± 3.14 μM.

Concerning other plant families, the leaves of *Annona muricata* are frequently used by Brazilians as a natural treatment of T2DM and its complications. Ethyl acetate fractions obtained from ethanolic extract of *A. muricata* leaves, which are rich in (poly)phenolic compounds, showed inhibition of alpha-amylase (IC_50_ 9.2 ± 2.3 μg/mL) and alpha-glucosidase (IC_50_ 413 ± 121 μg/mL), respectively, although less active than the ACB standard (IC_50_ 0.05 ± 0.01, for alpha-amylase, and 3.4 ± 0.6 μg/mL for alpha-glucosidase) [26].

Antidiabetic drugs aim mainly to achieve normoglycemia and minimize the progression of diabetes complications. Current antidiabetic drugs can interfere in one or more of the eight pathways which results in hyperglycemia. ACB, for instance, inhibits digestive enzymes, such as alpha-amylase, alpha-glucosidase, sucrase, and maltase. The inhibitor of these enzymes delays the digestion and absorption of dietary carbohydrates from the epithelium of the small intestine, and, thus, decreases the demand for insulin. Differently, STG acts to reduce the incretin effect in the small intestine by blocking the DPP-4 enzyme [13].

Due to the variety of mechanisms by which normoglycemia can be achieved, several tests can be performed to assess the antidiabetic potential of natural products. Generally, in vitro assays usually test the inhibition of digestive and DPP-4 enzymes. Our findings reveal that PESE and PIC can act as inhibitors of alpha-amylase, alpha-glucosidase, and DPP-4. Even though ACB and STG are more effective in terms of concentration in the inhibition of alpha-amylase and DPP-4, this extract and the pure compound PIC can be further investigated for use in natural antidiabetic formulations. Factors such as cost, safety, and side effects should be analyzed since the major concerns of the current drugs are the critical side effects associated with their over-use [13,15].

A recent study indicated that PIC inhibited the formation of *p*-nitrophenol by alpha-glucosidase through hydrophobic interactions and hydrogen bonding between hydroxy groups and amino acid residues in a non-competitive mechanism. The formation of the PIC-alpha-glucosidase complex may induce changes in the enzyme conformation, altering its function of digesting carbohydrates [27].

### 2.2. Inhibition of AGE in the Initial and Intermediate States of Glycation by the Extract

Glycation of BSA was performed in the presence and absence of PESE and PIC. Aminoguanidine (AMG) was used as a positive control. Inhibition of the initial and intermediate stages of glycation was carried out using a mixture of reducing sugars (fructose + glucose) and methylglyoxal (MGO), respectively.

The AGE formation inhibitory effect was proportional to the sample concentration. PESE and PIC were able to inhibit the formation of fluorescent AGEs for seven days of incubation at 37 °C with IC_50_ of 366 ± 1.9 and 51.5 ± 1.4 μg/mL for the BSA/fructose + glucose model, and 360 ± 9.1 and 67.4 ± 4.6 μg/mL for BSA/MGO, respectively (Table 2).

PIC showed higher activity when compared to PESE. The activity of PESE was only 2-fold lower than AMG in the initial stage of glycation and 1.3-fold lower in the intermediate stage. Therefore, PESE and PIC can act as antiglycation agents, probably by preventing the reaction of (di)carbonyl compounds from binding to protein and thus preventing the accumulation of AGEs.

In diabetes, AGEs are found in many organelles, associated with its complication, such as kidneys, retina, and atherosclerotic plaques. The inhibition of glycation is considered an effective strategy against the development or progression of degenerative diseases such as T2DM, atherosclerosis, and Alzheimer’s disease, and their complications, as well as in the reduction of chronic inflammatory processes [28,29].

The MGO trapping assay was performed employing its derivatization with *ortho*-phenylenediamine (OPD). PESE, PIC, and the positive control AMG presented 21.9%, 65.1%, and 99.6% of MGO trapping, respectively. Despite the above-shown differences, these results indicate that PESE is capable of acting in the intermediate stage of glycation, inhibiting reactive carbonyl species such as MGO and, consequently, it may reduce the formation of AGEs. The intermediate stage of the glycation reaction leads to the formation of reactive carbonyl species, such as MGO (a model compound), which in turn reacts with amino groups of biomolecules for the formation of AGEs [30]. Moreover, although the classic pathway of the Maillard reaction is well established as a trigger for the formation of AGEs, all reactions leading to the formation of α, β-dicarbonyl compounds in the body also contribute to the formation of AGEs [18]. Thus, the trapping capacity of MGO, a dicarbonyl compound produced during glycation, is an important step against the formation of AGEs.

Zhang, Wang, and Liu [31] observed that after reaction with MGO (5 mM), robinin, procyanidins, luteolin, quercetin, chrysoeriol, kaempferol, genistein, apigenin, and rutin at 5 mM showed capture percentage values of 54.4%, 46.3%, 40.7%, 21.4%, 41.8%, 58.9%, 43.4%, 36.5%, and 49.3%, respectively, values lower than those found in this work for PIC (65.1%, 2 mM). In another study, with ethanolic extract of onion peel (0.5 mg/mL), it was observed elimination of approximately 70% of MGO (0.33 mM) after 1 h of reaction, an excellent capture potential [32], while the methanolic extracts of berries and grapes, using a similar method, showed from 20% to 50% of MGO (10 mM) capture potential. Raspberry, strawberry, blackberry, cranberry, blueberry, and noble grape at a concentration of 2.5 mg/mL showed capture potential of 20%, 30%, 45%, 50%, and 50%, respectively, after 1 h of reaction with MGO [33].

### 2.3. Effect on the Formation of Amyloid Fibrils In Vitro

Protein glycation plays an important role in triggering or facilitating the aggregation of proteins, through the formation of crosslinked β structures that in turn leads to the formation of amyloid fibrils [34]. Thus, the potential of PESE and PIC for inhibiting the formation of amyloid fibrils was evaluated in two different systems, one with BSA in the presence of glucose and fructose (BSA/Fru + Glu) and the other in the presence of MGO (BSA/MGO). The thioflavin T (ThT) fluorescence assay was used as a general indicator for the formation of amyloid fibrils [35].

The formation of β-amyloid fibrils decreased as sample concentration increased for both systems, achieving 100% of inhibition, when treated with PIC at a concentration of 200 µg/mL (Table 2). AMG, however, was able to inhibit up to 35% of the formation of aggregates from the glycation reaction, being therefore less effective in maintaining the native structure of the protein, when compared to PESE and PIC. Our findings suggest that (poly)phenols of PESE might protect the protein backbone since the treatment with PESE and PIC decreased the formation of amyloid fibrils in BSA.

### 2.4. Antioxidant Activity

Our previous study identified six major phenolic compounds of PESE, including PIC, astringin, scirpusin A, scirpusin B, isookanin-7-*O*-glucoside, and naringenin-7-*O*-glucoside. Among them, PIC has emerged as a promising compound with important biological activities [36]. Hence, PIC was also investigated in this study, along with PESE.

The TPC of PESE corresponds to 227 mg GAE/g, dry weight (DW). DPPH^•^ and HOCl scavenging assays of PESE and PIC revealed that the antioxidant capacity of PESE is lower than that of PIC; however, this is similar to quercetin (QCT) in the DPPH^•^ scavenging assay (Table 3). HOCl scavenging activity showed that both PESE and PIC exhibit a strong antioxidant effect in a concentration-dependent manner; PIC had the highest activity and PESE was similar to QCT. Regarding the capacity of scavenging O_2_^•−^, PIC presented the highest potential followed by QCT and PESE.

Therefore, the results suggest that PIC may be an important (poly)phenolic compound in passion fruit seeds extracts. Furthermore, these results indicate that PESE is a potential antioxidant mixture against ROS. Recently, Rotta et al. [37] obtained similar results (250 ± 20 mg GAE/g, DW) for TPC and IC_50_ (19 ± 3 μg/mL) for DPPH^•^ in extracts of *P. edulis* seeds prepared by using ethanol:water solution (70:30, *v*/*v*) at 80 °C for 4 h.

Inhibiting intracellular ROS formation would provide a therapeutic strategy to counteract oxidative stress and cell damage [38]. Several in vitro and animal models and limited human studies have revealed that supplementation with a (poly)phenol-rich diet has attenuated oxidative stress, reduced postprandial and fasting blood glucose levels, and improved insulin release and sensitivity [39]. Therefore, antioxidant phytochemicals present in PESE, such as PIC, can contribute to homeostasis.

### 2.5. The Extract Reduces Intracellular ROS-Induced by a Carcinogen

To evaluate the pro-oxidant and/or antioxidant effects of PESE and PIC, cultured BEAS-2B, AML-12, and MCF-10A cells were exposed to a carcinogen alone or pretreated with PESE and PIC [40]. We have demonstrated that 100 µM of 4-[(acetoxymethyl) nitrosamine]-1-(3-pyridyl)-1-butanone (NNKOAc) induces intracellular ROS production in BEAS-2B cells without reducing cell viability [41]. NNKOAc is a precursor of NNK, which is the most abundant *N*-nitrosamine and the most potent carcinogen in cigarette smoke [42]. NNKOAc generates similar cytosolic reactive electrophilic compounds as NNK, which binds to DNA, forming DNA adducts and inducing DNA damage [41].

It was observed that PESE and PIC at 50 µg/mL were able to protect all cultured cells from the oxidative stress caused by the carcinogen (Figure 1). PESE at 100 µg/mL also prevented the ROS generation in BEAS-2B cells with NNKOAc. However, they were not able to prevent oxidative stress in AML-12 and MCF-10A cells. Our results demonstrate that the PESE and PIC can mitigate ROS generation, especially under oxidative stress, which is considered the primary event in diabetes.

It has long been recognized that hyperglycemia induces the formation of ROS, including superoxide radical anion (O_2_^•−^), hydroxyl radical (^•^OH), and hydrogen peroxide (H_2_O_2_). Under aerobic conditions, ROS are generated during normal metabolic activities, but the imbalance between the excessive ROS generation and lack of antioxidant defense are the primary factors that might cause lipid peroxidation, protein denaturation, and DNA mutation in healthy cells. These reactive species deteriorate β-cell function and increase insulin resistance, which triggers the aggravation of T2DM [29]. Treatment with PESE and PIC at lower concentrations might protect cells against glucose toxicity through scavenging ROS.

### 2.6. Cytotoxicity of the Seed Extract of P. edulis (PESE) and PIC

The examination of the cytotoxicity of natural extracts helps in the evaluation of their safety, which is important in the development of nutraceuticals or dietary supplements. When the cell viability is greater than 90%, a sample is nontoxic at the tested dose [43]. A dose-responsive decline in the viability of BEAS-2B, AML-12, and MCF-10A cells was observed with increasing PESE and PIC concentrations (*p* ≤ 0.05) (Figure 2). However, PESE did not impact the viability of BEAS-2B cells up to 100 µg/mL, while in AML-12 and MCF-10A cells, their viabilities were reduced by 25%. Up to 250 µg/mL PIC, the viability of AML-12 cells was maintained while in MCF-10A cells, viability was maintained above 80% up to 100 µg/mL PIC.

BEAS-2B cells appear to be more sensitive to PIC, with a reduction of approximately 50% of cell viability at the concentration of 100 µg/mL (409 µM). Despite the reduction in viability in some of the concentrations in the different cells studied, PESE and PIC do not reduce the cell viability in the bioactive concentrations that were effective in the inhibition of alpha-amylase, alpha-glucosidase, DPP-4, and antiglycation.

Hyperglycemia is well documented to have long-term effects on multiple organs, including the kidney, heart, brain, liver, and eyes. Diabetes also affects lung function by inducing inflammatory processes and fibrotic changes in the tissue [28]. Here, our results suggest that PESE and PIC are relatively safe for normal liver and lung cells and thus may not be harmful to organs and tissues. However, extensive toxicological assessments using experimental animal models need to be performed.

The cytotoxicity of PESE was investigated in the Vero E6 cell line and no decrease in cell viability was observed; however, at the highest concentration (100 μg/mL) there was an increase in cell viability, which according to the authors [36] may be related to a possible increase in mitochondrial proliferation or enzyme activity. The cell viability of human placental HTR-8/SVneo cells was also assessed in the presence of the extract, also with no reduction in cell viability at up to 100 μg/mL [36].

Yepes and colleagues have reported that the ethanol extract of purple passion fruit seeds at 1000 and 4000 µg/mL concentrations did not decrease the viability of normal human leukocyte cells, which is in contrast with the results of the present study [44]. Another study stated that an extract of defatted yellow passion fruit seed obtained using pressurized liquid extraction significantly decreased viability in all prostate cancer cell lines (PC-3, 22Rv1, LNCaP, and VcaP) in a dose- and time-dependent manner (10, 20, and 30 µM) [45].

## 3. Material and Methods

### 3.1. Chemicals

The analytical solvents and chemicals used for antioxidant and antiglycation activities were purchased from Sigma-Aldrich (Steinheim, Germany): Folin–Ciocalteu reagent (FC), DPPH^•^, β-nicotinamide adenine dinucleotide (NADH), 4-nitro blue tetrazolium chloride (NBT), *N*-methylphenazonium methyl sulfate (PMS), AMG, sodium hypochlorite solution (NaOCl), dihydrorhodamine 123 (DHR), QCT, OPD, and ThT. PIC was obtained from AK Scientific (Union City, CA, USA). The analytic solvents, chemicals, and enzymes used for antidiabetic assays were obtained from Sigma-Aldrich (Oakville, ON, Canada): pancreatic alpha-amylase, yeast alpha-glucosidase, ACB, STG, 2-chloro-4-nitrophenyl-α-D-maltotrioside, 4-nitrophenyl-α-D-glucopyranoside, trisodium phosphate, *N*-(2-hydroxyethyl)piperazine-*N*′-(2-ethanesulfonic acid), 4-(2-hydroxyethyl)piperazine-1-ethanesulfonic acid (HEPES), and DPP-4. All the reagents were of analytical grade, and the stock solutions and buffers were prepared with Milli-Q purified water. LHC-9 growth medium for BEAS-2B cells was purchased from ThermoFischer 91 (Chelmsford, MA, USA). MEBMTM mammary epithelial basal medium and supplementation (MEGMTM Mammary Epithelial Cell Growth Medium Kit) for MCF10-A cells, and DMEM F-12 medium (ATCC) supplemented with insulin-transferrin-selenium (ITS-G) for AML-12 cells were purchased from Lonza (Basel, Switzerland). NNKOAc was purchased from Toronto Research Chemicals (Toronto, ON, Canada). MGO, 3-(4,5-dimethylthiazol-2-yl)-5-(3-carboxy methyl phenyl)-2-(4-sulfophenyl)-2H-tetrazolium (MTS), phenazine methosulfate (PMS), and 2′,7′-dichlorofluorescein diacetate dye (DCFH-DA) were purchased from Sigma-Aldrich (Oakville, ON, Canada). Stock solutions were prepared with DMSO, and the final concentrations were not exceeding 0.5% (*v*/*v*) in the culture treatment medium.

### 3.2. Plant Material

The *P. edulis* material used in this study was collected in 2019 from the company “Polpa de Frutas Santa Luzia”, a local fruit processing industry located in Marechal Deodoro, Alagoas Brazil. Firstly, the remaining pulp in the seeds was removed manually. Seeds were washed in distilled water, dried in an oven at 50 °C for 48 h, and ground in a blender. The seed samples were kept in an amber flask at room temperature. Next, extraction was performed in a Soxhlet apparatus using 12 g of dry ground seeds. n-hexane (200 mL) was used to remove fat from the samples. Then, extraction was carried out with 200 mL ethanol for 6 h. Both solvents were removed by reduced pressure with a rotary evaporator. Extracts were maintained at 4 °C until further analysis. In this study, the ethanolic extract of *P. edulis* seeds (PESE) was evaluated.

### 3.3. Antidiabetic Activity Assays In Vitro

#### 3.3.1. Alpha-Amylase Inhibition Assay

Plant extract screening for alpha-amylase inhibition was conducted using the previously described method [46], with some modifications. Except as otherwise indicated, solutions were prepared in 0.01 M potassium phosphate buffer containing sodium chloride (60 mM) and sodium azide (0.02% *w*/*v*), pH 6.8. The PESE and PIC solutions were prepared in a buffer containing 8% ethanol. To a 96-well clear plate, 20 µL of plant extract and 20 µL of alpha-amylase from porcine pancreas (4 U/mL in buffer) were added. After 10 min of incubation at 37 °C, 20 µL of the substrate 2-chloro-4-nitrophenyl-α-D-maltotrioside (5 mM) was added. The mixture was then incubated for 30 min at 37 °C for the reaction to take place. The reaction was terminated by adding 240 µL of trisodium phosphate solution of pH 11 (1% *w*/*v*) to stop enzyme activity. Then, the absorbance at 405 nm was recorded using the microplate reader (Infinite^®^ 200 PRO, TECAN, Männedorf, Switzerland) to quantify the amount of 2-chloro-4-nitrophenol released. The effectiveness of the tested inhibitors was compared with ACB, a prescribed antidiabetic drug for alpha-amylase inhibition that was used as a means of comparison. The positive control was a mixture of enzyme and substrate without inhibitors. Sample control containing sample and buffer was used to eliminate color interference. Buffer was used as blank. The percentage of the alpha-amylase inhibition was calculated as I% = [(Abs_0_ − Abs_1_)/Abs_0_] × 100, where A_0_ is the absorbance of the positive control subtracted from the blank, Abs _1_ is the absorbance in the presence of the extract subtracted from sample control. The IC_50_ (half-maximal inhibitory concentration) was calculated graphically, using an analytical curve by plotting the concentration versus the inhibition percentage (I%).

#### 3.3.2. Alpha-Glucosidase Inhibition Assay

The alpha-glucosidase inhibition assay was performed using a previously described method [15] with minor modifications. First, solutions were prepared in 0.01 M potassium phosphate buffer (pH 6.8), unless otherwise stated. Various concentrations of PESE and PIC were prepared in a buffer containing 2.5% ethanol. To a 96-well clear plate, 120 µL of the samples and 20 µL of alpha-glucosidase enzyme (0.25 U/mL) were added. The plate was incubated at 37 °C for 15 min before adding 20 µL of 4-nitrophenyl-α-D-glucopyranoside (5 mM) substrate. The mixture was then incubated at 37 °C for 15 min for the reaction to take place. After incubation, the reaction was stopped by adding 80 µL of 0.2 M sodium carbonate in 0.1 M potassium phosphate buffer, pH 6.8. The absorbance at 405 nm was recorded using a microplate reader (Infinite^®^ 200 PRO, TECAN, Männedorf, Switzerland) to quantify the amount of *p*-nitrophenol (PNP) released. ACB, an antidiabetic drug used as an alpha-glucosidase inhibitor for T2DM, was used for comparison purposes. The positive control was the mixture of enzyme and substrate without inhibitors. The sample control consisted of the mixture of sample and buffer. The buffer was used as a blank. The percentage of the alpha-glucosidase inhibition and IC_50_ was calculated as described in Section 3.3.1.

#### 3.3.3. Dipeptidyl Dipeptidase Enzyme (DPP-4) Inhibition Assay

The DPP-4 inhibition assay was performed according to an established method [46]. Briefly, to a 96 well plate, 20 µL of the sample at different concentrations, 20 µL DPP-4 human recombinant enzyme solution (3.125 mU), and 50 µL of Gly-Pro-7-amido-4-methylcoumarin hydrobromide substrate (2.5 µM) were added. The reaction mixture was incubated for 30 min in the dark at 37 °C. Then, the fluorescent product was recorded at excitation/emission wavelengths of 350/450 nm using the microplate reader (Infinite^®^ 200 PRO, TECAN, Männedorf, Switzerland). STG, a standard DPP-4 inhibitor, was used to compare the effectiveness of PESE and PIC. The percentage of the DPP-4 inhibition and IC_50_ was calculated as described in Section 3.3.1.

### 3.4. Antiglycation Activity Assays In Vitro

#### 3.4.1. Inhibition of Advanced Glycation End Products (AGE) Formation in the Initial Stage of Glycation

The formation of AGEs was measured after incubation of a system containing protein and carbohydrates. This assay was based on previous methods [47]. The reaction system was obtained by adding 300 µL of plant extract or of the pure compound (at different concentrations), 150 µL D-fructose (200 mM), 150 µL of D-glucose (200 mM), and 300 µL of bovine serum albumin (BSA, 3 mg/mL) to a test tube. All these solutions, except plant extract, were dissolved in 0.05 M potassium phosphate buffer (pH 7.4) containing NaCl (100 mM) and NaN_3_ (0.02% *w*/*v*). The plant extract was prepared in a buffer containing 60% ethanol. The mixture was incubated in the dark at 37 °C for 7 days with constant stirring. After the incubation, 200 µL of the reaction mixture was transferred to a 96-well black plate, and fluorescent AGEs were measured using a microplate reader (Infinite^®^ 200 PRO, TECAN, Männedorf, Switzerland) at *λ*ex = 360 and *λ*em = 440 nm. X. Aminoguanidine (AMG), a known AGE formation inhibitor, was used as a standard. The reaction mixture without inhibitors was used as a positive control. Color interference from the sample was eliminated by a sample control (buffer and sample). The fluorescence readings for the experimental treatment were blanked against extract blanks to eliminate the baseline fluorescence of the sample. The percentage of the AGE inhibition and IC_50_ was calculated as described in Section 3.3.1.

#### 3.4.2. Inhibition of AGE Formation in the Intermediate Stage of Glycation

The BSA-MGO assay was based on those described before [33], with some modifications. This experiment is based on the formation of fluorescent AGEs from the middle stage of protein glycation. MGO (1.5 mM) and BSA (3 mg/mL) were dissolved separately in 0.05 M phosphate buffer (pH 7.4) containing NaCl (100 mM) and NaN_3_ (0.02% *w*/*v*). Then, 300 μL of MGO was incubated with samples at different concentrations (dissolved in a buffer containing 60% ethanol) for an hour at 37 °C, in the dark, with constant stirring. After BSA (3 mg/mL) was added to the reaction mixture and incubation time was extended to 48 h at 37 °C in darkness. The rest of the procedure was the same as that for the BSA-glucose/fructose model that is described above. The percentage of the AGE inhibition and IC_50_ was calculated as described in Section 3.3.1.

#### 3.4.3. Inhibition of the Formation of Amyloid Fibrils Using ThT Assay

The fibrillar state of modified BSA was determined with ThT as previously described [48]. ThT is a probe that binds directly to amyloid fibrils to give a strong fluorescent signal and thus can be used to quantify β-amyloid [35] due to protein glycation. Basically, 30 μL of glycated protein (approx. 1 mg/mL), obtained as described previously, was mixed with 100 μL ThT reagent (10 μM ThT in 100 mM phosphate buffer, pH 7.0), and fluorescence was collected at excitation/emission wavelengths of 450/490 nm in a microplate reader (Infinite^®^ 200 PRO, TECAN, Männedorf, Switzerland). Fresh BSA solution (1 mg/mL) and PBS were used as controls.

#### 3.4.4. Evaluation of MGO Capture Using Derivatization with OPD

MGO scavenging was tested using a described procedure [33], with slight modifications. The quantification of MGO was based on derivatization with OPD, leading to the formation of the product 2-methylquinoxyline (2-MQ). MGO and OPD were dissolved in phosphate buffer (50 mM, pH 7.4) to a concentration of 2 mM and 4 mM, respectively. AMG (2 mM) was used as a positive control, PESE and PIC were prepared with a concentration of 2 mg/mL and 2 mM, respectively. The reaction system was composed of 125 μL of the MGO solution, 125 μL of phosphate buffer (negative control), or PESE incubated at 37 °C for 1 h. After 250 μL of the OPD solution was added, the tubes were kept for 30 min for the derivatization reaction between MGO and OPD to complete. Then, the chromatographic analysis was performed. All solutions were previously filtered (microfilter pore diameter 0.45 μm). The conditions for the analysis by high-performance liquid chromatography (HPLC) were: deionized water with formic acid (0.1% *v*/*v*, solvent A) and methanol (solvent B) were used as mobile phases, the flow rate of 1.0 mL/min, and the injection volume was 20 μL. The linear gradient for elution was: starting at 5% of solvent B, 0–3 min, 5 to 50% B; 3–16 min, isocratic in 50% B; 16–17 min, 50–90% B; 17–19 min, isocratic in 90% B and 19–19.5 min, 90–5% B. The derivatization product, 2-methylquinoxyline (2-MQ), was detected at 315 nm, with a retention time of 10 min. The percentage of trapping efficiency of MGO was calculated using the following equation: % trapping MGO = [100 − (peak area after adding sample/peak area without adding sample) × 100].

### 3.5. Scavenging of Free Radical and Reactive Oxygen Species (ROS) Assays

#### 3.5.1. Total Phenolic Content (TPC)

The TPC was estimated using the Folin–Ciocalteu (FC) method, as described [49], with some modifications. Briefly, 180 μL of deionized water, 300 μL of FC reagent, and 2.4 mL of 5% sodium carbonate (*w*/*v*) were added to 120 μL of diluted samples. After incubation in a water bath at 40 °C in the dark for 20 min, the absorbance of the resulting mixture was measured at 760 nm using a UV–Vis spectrophotometer (Agilent 8453). The results were expressed as milligrams of gallic acid equivalents (mg GAE) per gram of dry extract.

#### 3.5.2. Radical Scavenging Assay DPPH^•^

Antioxidant activity of PESE was determined using the DPPH^•^ method [50]. Briefly, aliquots of 0.30 mL of sample dissolved in ethanol (5–25 μg/mL) were mixed with 2.70 mL of DPPH^•^ solution (40 μg/mL in methanol). After incubation in the dark for 30 min, the absorbance was read at 516 nm, using a UV–Vis spectrophotometer (Agilent Technologies, Santa Clara, CA, USA). Results were expressed as the half-maximal inhibitory concentration (IC_50_) in μg/mL.

#### 3.5.3. Hypochlorous Acid (HOCl) Scavenging Activity

HOCl scavenging activity of PESE and PIC was determined [51]. Briefly, a freshly prepared HOCl solution, by adjusting the pH of a 1% solution of NaOCl to 6.2 with dropwise addition of diluted H_2_SO_4_, was diluted to 30 µM, using 100 mM phosphate buffer pH 7.4. For the analysis, in a 96-well plate, the following reagents were added at the indicated final concentrations: 150 μL of buffer solution (100 mM, pH 7.4), 50 μL of PESE (1, 5, 10, 25, 50, 100, 200, and 300 µg/mL), 50 µL of DHR 123 (5 µM) and 50 µL of HOCl (5 µM). QCT was used as means of comparison. Fluorescence assays were performed in a microplate reader (Infinite^®^ 200 PRO, TECAN, Männedorf, Switzerland), at 37 °C, at wavelengths of 505 ± 10 nm and 530 ± 10 nm, for excitation and emission, respectively. The results were expressed as IC_50_ (μg/mL) of extract solution.

#### 3.5.4. Superoxide Anion Radical Scavenging Activity

The potential of the samples to scavenger superoxide anion radicals was determined [51], with minor modifications. In a 96-well plate, the following solutions were added to the final concentrations indicated: 50 μL of PESE (1 to 300 µg/mL), 50 μL of NADH (166 μM), 150 μL of NBT (43.3 μM), and 50 μL of PMS (2.7 μM). Phosphate buffer (19 mM, pH 7.4) was used to dissolve NADH, NBT, and PMS. QCT was used as means of comparison. The experiment was conducted at 37 °C in a microplate reader (Infinite^®^ 200 PRO, TECAN, Männedorf, Switzerland), and the absorbance was measured at 560 nm. The results were expressed as IC_50_ of extract solution in μg/mL.

#### 3.5.5. Measurement of Intracellular ROS Level

The generation of intracellular ROS in BEAS-2B, AML-12, and MCF-10A cells after treatments was measured as described previously [40]. Briefly, DCFH-DA dye was readily taken up by cells and is subsequently hydrolyzed to DCFH, which can be oxidized to a measurable fluorescent product dichlorofluorescein (DCF). The cells, pre-treated with PESE or PIC for 3 h, were exposed to the carcinogen NNKOAc, for 3 h, or alone in different experimental groups. Cells with DMSO media (0.5%) served as the vehicle control. After treatments, DCFH-DA was added to the cell culture plates at a final concentration of 5 μM followed by 40 min incubation at dark. The fluorescence degradation was then measured at an excitation and emission wavelength of 485 nm and 535 nm, using a microplate reader (Infinite^®^ 200 PRO, TECAN, Männedorf, Switzerland).

### 3.6. Cell Cultures and Cell Viability Assay

BEAS-2B, MCF10-A, and AML-12, cells were purchased from the American Tissue Type Culture Collection (ATCC; CRL-9609, CRL-10317, and CRL-2254) and cultured with a specific medium. BEAS-2B cells were cultured with LHC-9 media at 37 °C in an incubator with 5% CO_2_. Culture flasks (polystyrene T75) were pre-coated with a mixture of 0.01 mg/mL fibronectin, 0.03 mg/mL bovine collagen type I, and 0.01 mg/mL bovine serum albumin were dissolved in LHC-9 medium overnight. MCF10-A cells were cultured in mammary epithelial basal medium (MEBM) culture medium supplemented with the Mammary Epithelial Cell Growth Medium Kit (Lonza), and AML-12 cells in DMEM-F12 medium (ATCC) were supplemented with insulin-transferrin-selenium (ITS, GIBCO), 10% fetal bovine serum, and 1% antibiotic (penicillin-streptomycin, Gibco) at 37 °C in a 5% CO_2_ incubator. Cells were grown to around 70% confluence on the culture flask, and passages (<10) were employed for all experimental conditions.

MTS Cell Titer 96™ aqueous cell proliferation assay was used to determine cell viability [41]. BEAS-2B, AML-12, and MCF-10A cells were treated with PESE and PIC at different concentrations to determine the non-cytotoxic dose. For that, 1 × 10^4^ cells were plated on a 96-well plate with a growth medium of 100 μL/well. After 24 h, cells were treated with PESE or PIC for an additional 24 h. Fifteen microliters of MTS reagent (with PMS) was then added to each well and incubated for 3 h in the dark. Absorbance was recorded at 490 nm using a microplate reader (Infinite^®^ 200 PRO, TECAN, Männedorf, Switzerland). For each experiment, cells with DMSO media served as control, and cells with only culture medium and MTS reagent served as the blank.

### 3.7. Statistical Analysis

All analyses were performed in triplicate (*n* = 3) with three independent studies and using Graph-Pad Prism software (GraphPad Software Inc., San Diego, CA, USA). Data were presented as the mean ± standard deviation (SD), and analyses of variance, one-way ANOVA, followed by Tukey test and *p* ≤ 0.05 were considered significant between experimental groups.

## 4. Conclusions

Passion fruit seeds, by-products of the juice industry, have the potential for use as a low-cost antioxidant and bioactive source for developing nutraceuticals and dietary supplements, for managing blood glucose levels, and consequently, reducing the progression of complications of T2DM. In this work, PESE showed a potent antidiabetic, antiglycant, and antioxidant potential without being toxic to BEAS-2B cells at bioactive concentrations. Furthermore, it was observed that PESE could protect some of the tested cultured cell lines from the oxidative stress caused by the presence of the chemical carcinogen, NNKOAc. Although it is still early to suggest the use of ethanolic extract from passion fruit seeds in the treatment of T2DM, this work demonstrates its potential dietary use to manage T2DM and glycation-associated complications. Further investigations on the efficacy and safety of PESE, using pre-clinical experimental animal models are encouraged and are underway.

## Figures and Tables

**Figure 1 molecules-27-04064-f001:**
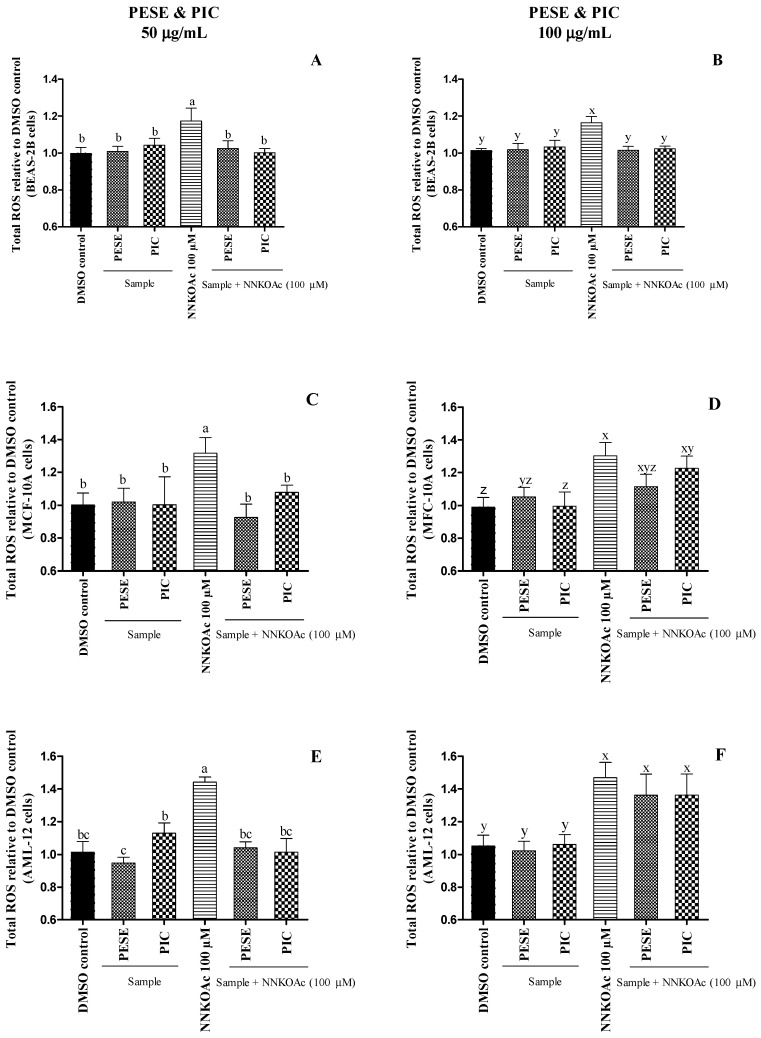
(**A**–**F**). The relative amount of reactive oxygen species (ROS) assessed on BEAS-2B, AML-12, and MCF-10A cell lines after exposure to either carcinogen alone or with pretreatment of PESE or PIC. All the treatment groups were compared to dimethyl sulfoxide (DMSO) control. a–c, x–z, mean ± SD followed by different letters represent significant differences (ANOVA analysis was performed followed by the Tukey test, *p* ≤ 0.05). Data are means of triplicates. Abbreviation: AML-12, alpha mouse liver 12; BEAS-2B, normal human bronchial epithelial cells; MCF-10, non-tumorigenic epithelial cells; DMSO, dimethylsulfoxide; NNKOAC, 4-[(acetoxymethyl)nitrosamine]-1-(3-pyridyl)-1-butanone; PESE, ethanolic extract of *P. edulis* seeds; PIC, piceatannol.

**Figure 2 molecules-27-04064-f002:**
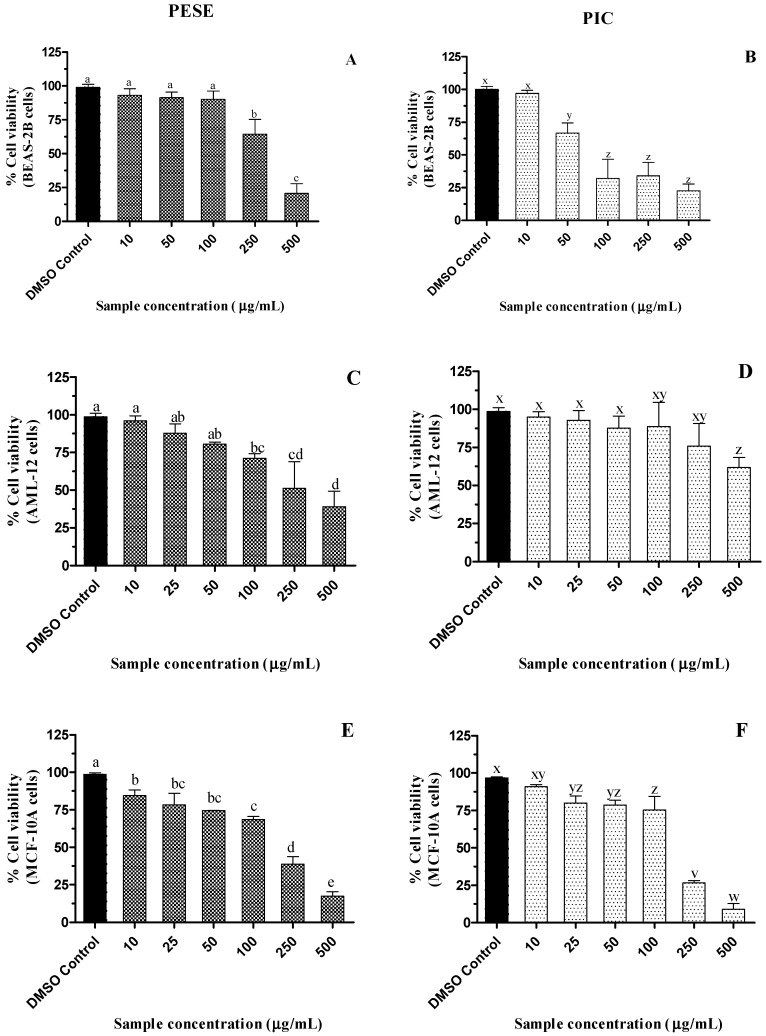
(**A**–**F**). Dose-dependent effect of ethanolic extract of passion fruit seeds on the viability of BEAS-2B, AML-12, and MCF-10A cell lines after 24 h of incubation. All the treatment groups were compared to dimethyl sulfoxide (DMSO) control. a–e, v–z, mean ± SD followed by different letters represent significant differences (ANOVA analysis was performed followed by the Tukey test, *p* ≤ 0.05). Data are means of triplicates. Abbreviation: AML-12, alpha mouse liver 12; BEAS-2B, normal human bronchial epithelial cells; MCF-10A, non-tumorigenic epithelial cells; DMSO, dimethylsulfoxide; PESE, ethanolic extract of *P. edulis* seeds; PIC, piceatannol.

**Table 1 molecules-27-04064-t001:** Antidiabetic activity in vitro of ethanolic extract of *P. edulis* seeds (PESE) and reference compounds.

Sample	Alpha-Amylase IC_50_ (μg/mL)	Alpha-Glucosidase IC_50_ (μg/mL)	DPP-4 IC_50_ (μg/mL)
PESE	32.1 ± 2.7 ^a^	76.2 ± 1.9 ^A^	71.1 ± 2.6 ^x^
PIC	85.4 ± 0.7 ^b^	20.4 ± 7.6 ^B^	1137.5 ± 120.2 ^y^
ACB	0.4 ± 0.1 ^c^	251.6 ± 4.5 ^C^	-
STG	-	-	0.005 ± 0.001 ^z^

IC_50_ values for alpha-amylase, alpha-glucosidase, and DPP-4 are given. ^a–c^, ^A–C^, ^x–z^, mean ± SD followed by different letters within the same assay represent significant differences (ANOVA analysis was performed followed by the Tukey test, *p* ≤ 0.05). Data are means of triplicates. Abbreviation: PIC, piceatannol; ACB, acarbose; STG, sitagliptin.

**Table 2 molecules-27-04064-t002:** The antiglycation activity and percentage of inhibition of formation of amyloid fibrils of ethanolic extract of *P. edulis* seeds (PESE) and reference compounds.

	Fructose + Glucose		MGO	
Sample	Antiglycation	Inhibition of Amyloid Fibrils (%)	Antiglycation	Inhibition of Amyloid Fibrils (%)
	IC_50_ (μg/mL)		IC_50_ (μg/mL)	
PESE	367 ± 1.9 ^a^	87.4 ± 2.7 ^b^	360 ± 9.1 ^a^	71.9 ± 4.5 ^b^
PIC	51.5 ± 1.4 ^b^	100 ± 5.3 ^a^	67.4 ± 4.6 ^b^	100 ± 3.8 ^a^
AMG	25.5 ± 5.0 ^c^	35.0 ± 5.9 ^c^	50.4 ± 1.8 ^c^	30.3 ± 4.4 ^c^

IC_50_ values were estimated using two models of glucose + fructose and methylglyoxal (MGO). The percentage values of inhibition of the formation of amyloid fibrils presented are the highest obtained for PESE, PIC, and AMG at concentrations of 200, 200, and 100 µg/mL, respectively. ^a–c^ Mean ± SD followed by different letters in the same assay represent significant differences (ANOVA analysis was performed followed by the Tukey test, *p* ≤ 0.05). Data are means of triplicates. Abbreviations: PIC, piceatannol; AMG, aminoguanidine; MGO, methylglyoxal.

**Table 3 molecules-27-04064-t003:** Total phenol content (TPC) and antioxidant capacity of ethanolic extract of passion fruit seeds and piceatannol, toward HOCl and O_2_^•−^.

Sample	TPC (mg GAE/g DW)	DPPH^•^ IC_50_ μg/mL	HOCl IC_50_ μg/mL	O_2_^•−^ IC_50_ μg/mL
PESE	227 ± 3.9	20.4 ± 2.1 ^a^	1.7 ± 0.3 ^a^	38.2 ± 0.5 ^a^
PIC	-	6.3 ± 1.3 ^b^ (25.8)	1.2 ± 0.5 ^b^ (7.8)	7.3 ± 0.02 ^c^ (30.1)
QCT	-	4.8 ± 1.0 ^b^ (15.9)	1.9 ± 0.3 ^a^ (4.0)	8.8 ± 0.3 ^b^ (29.2)

PESE, ethanolic extract of *P. edulis* seeds; PIC, piceatannol, QCT, quercetin, DW, dry weight; GAE, gallic acid equivalents; DPPH^•^, 1,1-diphenyl-2-picrylhydrazyl; HOCl, hypochlorous acid; O_2_^•−^, superoxide anion radical; ^a–c^, mean ± SD followed by different letters within the same column represent significant differences (ANOVA analysis was performed followed by the Tukey test, *p* ≤ 0.05). Data in parenthesis are in μM. Data are means of triplicates.

## Data Availability

Not applicable.

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
