# Peer review of "Antidiabetic, Antiglycation, and Antioxidant Activities of Ethanolic Seed Extract of Passiflora edulis and Piceatannol In Vitro"

_molecules, 2022, doi:10.3390/molecules27134064_

Round 1

Reviewer 1 Report

The authors suggest antidiabetic, antiglycation, and antioxidant activities of ethanolic seed extract of Passiflora edulis and piceatannol in vitro. The manuscript seems to be well written.

However, according to the previous reports, there have been several reports regarding antidiabetic, antiglycation, and antioxidant activities of P. edulis and piceatannol.

I think this manuscript seems just a combination of several things. In this regard, the author should explain the advantages of the present study compared to previous studies in the discussion or conclusion section. 

Author Response

Response of authors:

Thank you for the comment. We added the lines 98-102 at the end of the introduction to show the novelty of the present study. The antioxidant potential of the Passiflora edulis seeds had been reported before (as indicated in the introduction). However, this manuscript is beyond the antioxidant potential and describes many biological activities of seeds extracts of P. edulis for the first time.

Although there are reports on the antidiabetic and antioxidant activity of piceatannol, this compound was used in our work to verify if it would be responsible for the major activity of the ethanolic extract of the seeds of Passiflora edulis or if there are better results in the extract. Until now, although extensive reports on biological aspects of Passiflora edulis, there are, up to our knowledge, no studies with the extract of the seeds of the fruit as an antidiabetic agent through investigation of the inhibition of the following enzymatic panel: alpha-glucosidase, alpha-amylase and dipeptidyl-peptidase-4. There are also no data about the antiglycant potential in the initial and intermediate stages of glycation, as well as for inhibition of fibrillation of proteins subjected to glycation. There are, however, data related to the antioxidant potential, but we still add novel data regarding the elimination of HOCl and O2•-. To make our contribution to the literature evident, we added a paragraph at the end of the introduction showing the novelty of the present investigation.

Lines 96-99 No studies have been, up to our knowledge, reported with the seeds of Brazilian P. edulis (yellow passion fruit) as an antidiabetic agent through the association of alpha-glucosidase, alpha-amylase and DPP-4 inhibition mechanisms, not even as an antiglycant in the initial and intermediate stages of glycation and as an inhibitor of fibrillation of proteins subjected to glycation.

Reviewer 2 Report

The article presents very interesting results on the subject of analysis.

Abstract: IC50 values ​​for DPPH• radical scavenging activity of PESE and PIC were 20.4 ± 2.1, and 1.7 ± 0.3, respectively. Change for: IC50 values ​​for DPPH• radical scavenging activity of PESE and PIC were 20.4 ± 2.1, and 1.7 ± 0.3 µg/mL, respectively.

DPPH and HOCl must be defined correctly.

PESE, PIC, and acarbose (ACB) exhibited IC50 for alpha-amylase, 32.0 ± 2.7, 85.4 ± 0.7, and 0.40 ± 0.05 µg/mL, respectively. In the results it is found as (Lines: 108-109) PESE, PIC, and acarbose (ACB) exhibited IC50 for alpha-amylase, 32.0 ± 2.7, 85.4 ± 0.7, and 0.4 ± 0.05 µg/mL, respectively. Unify the criteria for the presentation of numerical figures and correct the unnecessary decimals set by Word or Excel.

Include the objective of the work in the abstract

Results and discussion:

There must be a balance between the number of figures and tables and this balance is not reflected in this work. Some of the figures can be transformed into a table.

In Figure 1A, B and C the internal marks of the bars must be differentiated. The three samples tested PESE, PIC and STG are represented by bars with internal lines. They could be exchanged for different bars. As well as this it seems that they are the same samples. Indicate in the legend of the figure that an ANOVA analysis was performed followed by the Tukey test.

Table 1 define PESE, PIC and QCT. Also indicate that an ANOVA analysis was performed followed by a Tukey test. DW, dry weight

Lines 126-132: Should be compared in the same units. Convert ug/mL to mg/mL so readers can easily compare data. When comparing data they should be in the same units if possible. If the candidates are of large or very small figures, it is understood that there are differences, but this is not the case. Correct throughout the manuscript.

Extract of passion fruit peel also displayed higher potential towards alpha-glucosidase than alpha-amylase. The ethanolic extract of a hybrid passion fruit peel had a weak inhibitory effect in alpha-amylase (1.8 ± 0.1 mg/mL) but a stronger effect in al pha-glucosidase (0.6 ± 0.1 mg/mL), which was 4.3-fold higher than standard drug ACB (0.1 ± 0.3 mg/mL) [23].

The same pattern was described by Loizzo et al. (2019), when investigating the hypoglycemic activity of ethanol extracts of seeds, peel, and pulp of ten Columbia native Passiflora species. Among all studied extracts, P. ligularis seeds + pulp had the greatest activity with IC50 values ​​of 22.6 and 24.8 μg/mL.

Line 242-243: The TPC of PESE corresponds to 227 mg gallic acid equivalent (GAE) per g dry extract ( ). Change for: The TPC of PESE corresponds to 227 mg GAE/g, DW.

Line 252: Rotta et al. (2020) obtained similar results (250 ± 20 mg GAE/g for TPC and 19 ± 3 µg/mL for DPPH•). Change for: Rotta et al. (2020) obtained similar results (250 ± 20 mg GAE/g, DW for TPC and IC50 (19 ± 3 µg/mL) for DPPH. Unify units ug/mL, mg/mL

In Figure 3 the control bar used in each assay should be distinguished from the samples. Because it can be confusing. Do not put the same internal drawing of the samples. Change for bars with a color without figures. For example, black bars.

Review all legends of tables and figures. All abbreviations must be defined and must be self-explanatory without requiring the text of the manuscript.

Line 385: ride (60 mM) and sodium azide (0.02% w/v),…Line 391: of pH 11 (1 % w/v) to. Correct formatting errors throughout the manuscript. Spaces between the % and °C symbols. Unify a criterion because both forms are found in the same paragraph.

Line 392: (Infinite® 200 PRO, TECAN, Switzerland) indicate the city.

Line 483: 3.5.1. Total phenol content. Change for 3.5.1. Total phenol content (TPC).

Line 484: The TPC was estimated…

Line 492: 3.5.2. DPPH• radical scavenging assay. Change for radical scavenging assay DPPH•

Line 493: Antioxidant activity of (???) was determined using the DPPH• method [50]…

The conclusions should not contain the results expressed in numerical figures. They have already been indicated in the abstract and in the results. It is repetitive so it is better to indicate it in another way.

In this work, PESE showed a potent antidiabetic, antiglycant, and antioxidant potential without being toxic to BEAS-2B cells at the concentration expected to display antidiabetic (up to 76.2 ± 1.9 µg/mL) and antioxidant (up to 38.2 ± 0.5 µg/ mL) potentials.

Best regards, 

Author Response

Reviewer 2:

Thank you for the comments.

The article presents very interesting results on the subject of analysis.

Abstract: IC50 values for DPPH• radical scavenging activity of PESE and PIC were 20.4 ± 2.1, and 1.7 ± 0.3, respectively. Change for: IC50 values for DPPH• radical scavenging activity of PESE and PIC were 20.4 ± 2.1, and 1.7 ± 0.3 µg/mL, respectively.

DPPH and HOCl must be defined correctly.

Response of authors: Corrections were made as below.

PESE, PIC, and acarbose (ACB) exhibited IC50 for alpha-amylase, 32.0 ± 2.7, 85.4 ± 0.7, and 0.40 ± 0.05 µg/mL, respectively. In the results it is found as (Lines: 108-109) PESE, PIC, and acarbose (ACB) exhibited IC50 for alpha-amylase, 32.0 ± 2.7, 85.4 ± 0.7, and 0.4 ± 0.05 µg/mL, respectively. Unify the criteria for the presentation of numerical figures and correct the unnecessary decimals set by Word or Excel.

Response of authors: We have made the changes by adding the unit µg/mL

Include the objective of the work in the abstract.

Response of authors:

We have included the objective of the work in the first sentence of the abstract. We unified the presentation of numerical figures with one decimal.

Results and discussion:

There must be a balance between the number of figures and tables and this balance is not reflected in this work. Some of the figures can be transformed into a table. In Figure 1A, B and C the internal marks of the bars must be differentiated. The three samples tested PESE, PIC and STG are represented by bars with internal lines. They could be exchanged for different bars. As well as it seems that they are the same samples.

Indicate in the legend of the figure that an ANOVA analysis was performed c.

Response of authors:

Figure 1A, B and C and Figure 2 have been transformed into tables. Also, it was indicated in the legend of each table that ANOVA analysis was performed followed by a Tukey test.

Table 1 define PESE, PIC and QCT. Also indicate that an ANOVA analysis was performed followed by a Tukey test. DW, dry weight

Response of authors:

PESE, PIC and QCT were defined in table 1 which is now table 2. Also, it was indicated in the legend of the table that ANOVA analysis was performed followed by a Tukey test. Abbreviation of DW was corrected.

Lines 126-132: Should be compared in the same units. Convert ug/mL to mg/mL so readers can easily compare data. When comparing data they should be in the same units if possible. If the candidates are of large or very small figures, it is understood that there are differences, but this is not the case. Correct throughout the manuscript.

Response of authors: All the corrections were made.

Extract of passion fruit peel also displayed higher potential towards alpha-glucosidase than alpha-amylase. The ethanolic extract of a hybrid passion fruit peel had a weak inhibitory effect in alpha-amylase (1.8 ± 0.1 mg/mL) but a stronger effect in alpha-glucosidase (0.6 ± 0.1 mg/mL), which was 4.3-fold higher than standard drug ACB (0.1 ± 0.3 mg/mL) [23].

The same pattern was described by Loizzo et al. (2019), when investigating the hypoglycemic activity of ethanol extracts of seeds, peel, and pulp of ten Columbia native Passiflora species. Among all studied extracts, P. ligularis seeds + pulp had the greatest activity with IC50 values of 22.6 and 24.8 μg/mL.

Response of authors: Conversion of units from the literature from mg/mL to ug/mL has been made.

Line 242-243: The TPC of PESE corresponds to 227 mg gallic acid equivalent (GAE) per g dry extract ( ). Change for: The TPC of PESE corresponds to 227 mg GAE/g, DW.

Response of authors: The change of “gallic acid equivalent (GAE) per g dry extract” for “GAE/g, DW” has been made.

Line 252: Rotta et al. (2020) obtained similar results (250 ± 20 mg GAE/g for TPC and 19 ± 3 µg/mL for DPPH•). Change for: Rotta et al. (2020) obtained similar results (250 ± 20 mg GAE/g, DW for TPC and IC50 (19 ± 3 µg/mL) for DPPH.

Unify units ug/mL, mg/mL

Response of authors: Thanks. The change has been made and IC50 was converted to mg/mL.

In Figure 3 the control bar used in each assay should be distinguished from the samples. Because it can be confusing. Do not put the same internal drawing of the samples. Change for bars with a color without figures. For example, black bars.

Response of authors: The internal drawing of the DMSO control has been changed to black bars.

Review all legends of tables and figures. All abbreviations must be defined and must be self-explanatory without requiring the text of the manuscript.

Response of authors: Thanks. Legends of tables and figures were reviewed.

Line 385: ride (60 mM) and sodium azide (0.02% w/v),…Line 391: of pH 11 (1 % w/v) to. Correct formatting errors throughout the manuscript. Spaces between the % and °C symbols. Unify a criterion because both forms are found in the same paragraph.

Response of authors: Thanks. Criterion used was no space between the number and %, space between number and °C.

Line 392: (Infinite® 200 PRO, TECAN, Switzerland) indicate the city.

Answer:(Infinite® 200 PRO, TECAN, Männedorf, Switzerland)

Line 483: 3.5.1. Total phenol content. Change for 3.5.1. Total phenol content (TPC).

Line 484: The TPC was estimated…

Line 492: 3.5.2. DPPH• radical scavenging assay. Change for radical scavenging assay DPPH•

Line 493: Antioxidant activity of (???) was determined using the DPPH• method [50]…

Response of authors: All suggested changes were made.

The conclusions should not contain the results expressed in numerical figures. They have already been indicated in the abstract and in the results. It is repetitive so it is better to indicate it in another way.

In this work, PESE showed a potent antidiabetic, antiglycant, and antioxidant potential without being toxic to BEAS-2B cells at the concentration expected to display antidiabetic (up to 76.2 ± 1.9 µg/mL) and antioxidant (up to 38.2 ± 0.5 µg/ mL) potentials.

Response of authors: The results expressed in numerical figures were removed. It is now: In this work, PESE showed a potent antidiabetic, antiglycant, and antioxidant potential without being toxic to BEAS-2B cells at bioactive concentrations.

Reviewer 3 Report

The manuscript deals with the interesting topic of in vitro antidiabetic potential of seed extract of Passiflora edulis and piceatannol. The study is well designed with an appropriate methodology. The paper is interesting and suitable for the scope of this journal. The manuscript is well designed with appropriate applied experimental methodology and is suitable for publication in the Molecules journal. However, I have some suggestions for correction of the manuscript before publication:

1.       The results for the formation of amyloid fibrils in vitro (Section 2.3.) should be presented in the picture or in the table. 

2.       Figures 3 and 4 appear to be identical. Described results for the effects of the extract on intracellular ROS induced by a carcinogen are not in accordance with the results presented in Figure 3.

3.       The procedure for calculating the IC50 for Alpha-amylase, Alpha-glucosidase, and DPP-4 inhibition assays is not described.

4.       Is there a possibility for the authors to perform an HPLC analysis of the extract (PESE) or to quantify PIC in the extract? This would significantly improve the quality of the manuscript.

Author Response

  1. The results for the formation of amyloid fibrils in vitro (Section 2.3.) should be presented in the picture or in the table. 

Response of authors: We have included the results in Table 2.

  1. Figures 3 and 4 appear to be identical. Described results for the effects of the extract on intracellular ROS induced by a carcinogen are not in accordance with the results presented in Figure 3.

Response of authors: Thanks for the comment. We have corrected this discrepancy and the correct Figure 3 is included.

  1. The procedure for calculating the IC50for Alpha-amylase, Alpha-glucosidase, and DPP-4 inhibition assays is not described.

 Response of authors: We have added the information to the text.

  1. Is there a possibility for the authors to perform an HPLC analysis of the extract (PESE) or to quantify PIC in the extract? This would significantly improve the quality of the manuscript.

Response of authors: Thanks for the comment.

The chemical identification of some of the compounds present in the extract was published in another paper (Anti-Zika Virus Effects, Placenta Protection and Chemical Composition of Passiflora edulis Seeds Ethanolic Extract. Doi: https://dx.doi.org/10.21577/0103-5053.20220003) and the quantification of piceatannol was not performed, at the time being. It is under way, but will require a longer time. As such, we apologize for not being able to give you a positive answer.

References were updated.

Round 2

Reviewer 1 Report

I think that the authors explain the merit of the present study in the introduction.

Reviewer 2 Report

Thanks to the authors for responding to the comments. I have no further comments on the work. The article can be accepted in its current form. 

Best regards,